# Chagas Heart Disease: Beyond a Single Complication, from Asymptomatic Disease to Heart Failure

**DOI:** 10.3390/jcm11247262

**Published:** 2022-12-07

**Authors:** Isis G. Montalvo-Ocotoxtle, Gustavo Rojas-Velasco, Olivia Rodríguez-Morales, Minerva Arce-Fonseca, Luis A. Baeza-Herrera, Arturo Arzate-Ramírez, Gabriela Meléndez-Ramírez, Daniel Manzur-Sandoval, Mayra L. Lara-Romero, Antonio Reyes-Ortega, Patricia Espinosa-González, Erika Palacios-Rosas

**Affiliations:** 1Cardiovascular Critical Care Unit, National Institute of Cardiology “Ignacio Chávez”, Juan Badiano No. 1, Col. Sección XVI, Tlalpan, Mexico City 14080, Mexico; 2Department of Molecular Biology, National Institute of Cardiology “Ignacio Chávez”, Juan Badiano No. 1, Col. Sección XVI, Tlalpan, Mexico City 14080, Mexico; 3Magnetic Resonance Imaging Department, National Institute of Cardiology “Ignacio Chávez”, Juan Badiano No. 1, Col. Sección XVI, Tlalpan, Mexico City 14080, Mexico; 4Academic Department of Health Sciences, School of Sciences, Universidad de las Américas Puebla, Ex Hacienda Sta. Catarina Mártir S/N. San Andrés Cholula, Puebla 72810, Mexico

**Keywords:** Chagas disease, Chagas cardiomyopathy (CC), chronic Chagas cardiomyopathy (CCC), myocarditis, heart failure

## Abstract

Chagas cardiomyopathy (CC), caused by the protozoan *Trypanosoma cruzi*, is an important cause of cardiovascular morbidity and mortality in developing countries. It is estimated that 6 to 7 million people worldwide are infected, and it is predicted that it will be responsible for 200,000 deaths by 2025. The World Health Organization (WHO) considers Chagas disease (CD) as a Neglected Tropical Disease (NTD), which must be acknowledged and detected in time, as it remains a clinical and diagnostic challenge in both endemic and non-endemic regions and at different levels of care. The literature on CC was analyzed by searching different databases (Medline, Cochrane Central, EMBASE, PubMed, Google Scholar, EBSCO) from 1968 until October 2022. Multicenter and bioinformatics trials, systematic and bibliographic reviews, international guidelines, and clinical cases were included. The reference lists of the included papers were checked. No linguistic restrictions or study designs were applied. This review is intended to address the current incidence and prevalence of CD and to identify the main pathogenic mechanisms, clinical presentation, and diagnosis of CC.

## 1. Introduction

American Trypanosomiasis, better known as Chagas disease (CD), described in 1909 by Dr. Carlos Justiniano Oswaldo Ribeiro das Chagas, has been a pathology recognized for its epidemiological significance in America and its current worldwide dissemination [1,2]. This pathology has had advances in diagnosis, control, and treatment in its early stages [3]. In 2005, the World Health Organization (WHO) considered it a “Neglected Tropical Disease” (NTD), and in 2020 established management objectives with targets for 2030, which included stopping congenital, vector-borne, transfusion-transmitted, and organ donation transmission, with a goal of 75% access to parasite treatment in all populations at risk [4]. Due to the context of the health and socioeconomic crisis associated with the COVID-19 pandemic, these advances and objectives have been heavily affected [4].

It is estimated that by 2025 CD will be responsible for 200,000 cardiovascular deaths worldwide, hence the importance of reviewing updated information on this pathology as a global health problem [5]. This article discusses the cardiovascular condition in the CD from its epidemiological and socioeconomic impact to its pathophysiology, risk factors, clinical manifestations, and diagnosis, as well as the latest advances in managing and preventing this disease.

## 2. Methods

The aim of this review was to summarize recent studies about the cardiovascular condition in Chagas disease (CD). For this purpose, a thorough literature review was conducted to obtain that satisfies our objectives.

### Literature Search

An extensive literature search of articles was conducted with the assistance of a medical librarian in six electronic databases (Medline, Cochrane Central, EMBASE, PubMed, Google Scholar, EBSCO) without any language restriction. Articles in non-English or non-Spanish were assessed using Google translation or PONS translation. The searches were restricted to January 1968, and the date of the final searches was October 2022. We searched combined terms related to “Chagas disease” AND “Chagas cardiomyopathy,” also with terms related to “electrocardiographic abnormalities,” “seroprevalence,” “vaccines,” “blood transfusion,” “immunology,” “image techniques,” and “nanomedicine.” Additionally, a manual search was performed for bibliographic references of the selected articles, and grey literature databases were also included to minimize publication bias (Figure 1).

## 3. Epidemiology and Socioeconomic Impact

Although CD is endemic to the Americas, it has undergone changes in its geographic distribution and epidemiology due to new waves of migration and global growth [2,6]. There are an estimated 6–8 million infected people worldwide, 65–100 million at risk of infection, and 28,000 new cases annually [5,7,8,9]. Chagas cardiomyopathy (CC) causes 10,000–14,000 deaths annually and is considered the leading cause of non-ischemic cardiomyopathy in Latin America [7,10,11,12].

In the 1990s, the WHO estimated ~18 million infected people in Latin America alone [1,13,14,15,16], not including the Caribbean islands [4], a figure that decreased by up to 70% as a result of vector control, spread containment, and blood product transmission programs in the Southern Cone [17]; however, it is still considered an NTD by the WHO.

The countries with the highest number of infected individuals are Brazil (1.9 million), Argentina (1.6 million), and Mexico (1.1 million) [18,19]. Bolivia (Bolivian Chaco) has the highest infection rate per year (4%), being the leading cause of non-ischemic cardiomyopathy in this region, affecting 7% of the working population [20].

Secondary to migration and constant population movement in recent decades, more and more cases are being documented in non-endemic regions such as the northern United States (U.S.), Canada, Europe, Asia, and Oceania, becoming a global health problem [7,18] (Figure 2). According to a cross-sectional study in the U.S. from 2002 to 2017, the number of hospitalizations for CC has increased significantly, with the highest admission rate being in western and southern U.S. regions, the areas of highest migration from Latin America [21]. However, it is currently debatable to infer that CC and CD incidence in the U.S is solely associated with the Latin America migration process. In recent years, an increase in autochthonous CD has been described, which has led to different questions and rethinking of new pathways of transmission [22].

Demographic studies showed migration patterns from South America to countries such as the U.S., with an estimated prevalence of ~300,000 infected inhabitants [21,23,24,25]**,** of which 30 to 45 thousand cases corresponded to undiagnosed CC [24,25]. In the European population, the estimated prevalence of CD is 123,000 people affected [25] in countries such as Spain, Italy, and Switzerland, which are places of major migration from Latin America [26]. This meta-analysis studied a group of 10,000 Latin American immigrants residing in Europe, in which the prevalence of CD was 4.2% in Bolivians and Paraguayans [26].

The re-emergence of CD and migratory movements to non-endemic areas lead to significant economic losses in endemic countries and those where the disease is emerging. In most cases, these are detected in chronic stages affecting economically active people, in which the only treatment will be symptomatic control as well as that of complications, resulting in an “incapacitating disease” [3,8]. In 1991, an approximate annual loss of 1819 accumulated working years per 100,000 infected inhabitants was estimated, equivalent to 1208 million dollars (USD) per capita gross national income in countries such as Bolivia, Chile, Paraguay, Peru, and Uruguay, with an investment of 2000 to 4000 USD per person, only for the care of the after-effects of CD [13].

In more recent estimates, and according to the model proposed by Lee et al. (2013), CD reduces 0.51 disability-adjusted life-years (DALYs), with an individual cost of 474 USD annually, and globally it accounts for 627,460,000 USD in health care expenditures and a loss of 382,250 DALYs [27]. The interpretation of these data is alarming, as it can be compared to the economic burden of diseases such as rotavirus (2.0 million USD), cervical cancer (4.7 million USD), and Lyme disease (2.5 million USD in the U.S. alone), indicating the need to implement stricter control and prevention measures [27].

The main problems identified for containing and decreasing the socioeconomic impact of CD can be summarized as follows: (a) limited diagnosis and clinical follow-up; (b) underfunding and poor organization in the care of patients without access to a universal health system; (c) lack of knowledge on the part of primary care physicians and non-medical health personnel and (d) lack of financial resources for education and research [23].

In addition, National Health Systems focused on low- and middle-income Mexican people are not effective because: (i) patient care requires a lot of bureaucratic procedures, (ii) the time between one medical consultation and another is very long, which makes follow-up is delayed a lot, (iii) sometimes there are no tools (supplies and equipment) for diagnosis or treatment, and there is also no trained personnel to perform these interventions or interpretation, (iv) although the diagnosis exists, sometimes it is not official, since validation by the corresponding health jurisdiction is required, (v) despite having a validated diagnosis, treatment is not available, so all this has an impact on the progression of the disease, and in the future, the patient loses years of productive life due to a disabling disease. Patients unable to work become an economic burden for their families, for health systems, and at the national level, causing a decrease in the productive rates of the economically active population.

## 4. Pathophysiology of Chagas Cardiomyopathy

### 4.1. Biology and Transmission Routes of Trypanosoma cruzi

The protozoan *Trypanosoma cruzi* causes CD, and it can infect vertebrate and invertebrate hosts, coupled with the ability to survive intracellularly and in the circulatory system [8,28,29,30]. This protozoan was first discovered and described by Dr. Carlos Ribeiro Justiniano Oswaldo das Chagas in 1909 [29,31,32]. Hematophagous triatomine bugs, better known as kissing bugs, are responsible for two-thirds of the infections [2,12,20]. To date, 150 species of the subfamily Triatominae capable of transmitting *T. cruzi* have been recognized, which are endemic in some regions of the Americas [1,8,13,19,33,34,35]. The other forms of transmission of the disease are maternal-fetal transmission (26%) and transmission associated with the transplantation of infected organs and blood products (~1%) [7,9,36].

The life cycle of *T. cruzi* is complex and varied and occurs in two phases: that of the vector and that of the mammal (including humans) (Figure 3) [8,9,20,30,32]. During this period, the parasite will evolve through three main morphological and functional stages: trypomastigote, epimastigote, and amastigote [37].

CD is more common in children under five years of age and has no gender preference [12]; however, the World Health Federation [7] emphasizes that the current trend is focused on male adults and older people with risk factors [41]. In turn, *Trypanosoma cruzi* transmission can be secondary to contaminated food (acute outbreaks in Brazil and the Amazon region with higher mortality in the acute phase) [1,7,8,9,12,39].

### 4.2. Genetics

*T. cruzi* has a broad genetic architecture that changes as it adapts to its environment, characterized by a diploid genome with a high degree of polymorphism among its homologous chromosomes [28,42,43,44]. This kinetoplastid protozoan is currently divided into seven discrete typing units (DTU) or lineages called TcI-TcVI and Tcbat (this is described in bats) [2,42].

A relationship has been established between the different types of DTU with the type of geographical area, type of host (vertebrates, invertebrates), and pathogenicity [2,45,46]. For example, TcI is the most diverse and widespread DTU in Latin America, being the main cause of CD in Central America and Southern Cone countries [2,45]. Conversely, DTU TcII, TcV, and TcVI are found in the Amazonian areas and are considered the most representative of South America [2,45].

There are differences in clinical manifestations, virulence patterns, biological processes, and molecular functions among the different strains of *T. cruzi* belonging to the various lineages [28,42]. The most widely used strain in research and diagnostics is Y (DTU II) [47]. Proteomic studies in BALB/c mice [42] and non-isogenic Swiss albinos [47] have evidenced a relationship between parasitemia and acute phase lethality [42].

In addition, the constant change between hosts (vertebrate to invertebrate, and vice versa) has resulted in morphological and immunological adaptations [44,46]. Changes at the microRNA level (such as post-transcriptional deregulation) associated with cardiovascular disease and present in cardiac remodeling in CC have been discovered and are being studied for new pharmacological therapies [47,48].

### 4.3. Pathogenesis

The pathophysiology pathogenesis of CC is complex and involves multiple pathways and systems that interact with each other and can be encompassed in four major groups: (1) parasite-dependent myocardial damage, (2) parasite-independent myocardial damage, (3) neurogenic disorders, and (4) microvascular disorders (Figure 4) [49].

#### 4.3.1. Parasite-Dependent Myocardial Damage

*T. cruzi* can infect myocytes, endothelial cells, fibroblasts, and adipocytes at the cardiovascular level; however, it has a tropism for cardiac muscle tissue [50,51]. During the acute stage, cardiac damage is related to the presence of the parasite and is caused by mechanical rupture of the infected cell, the release of waste, and attraction of inflammatory cells or by some toxic product released by the microorganism –acid-active hemolysin (TC-TOX) and LYT1 [2,9,40,52]. These molecules induce myocytolysis, which occurs following the differentiation of intracellular amastigotes into blood trypomastigotes; all this leads to important cardiac damage. Moreover, TC-Tox and LYT1 are immunologically related to the human complement protein C9 and have similar hemolytic activity [40,53].

In 90% of patients with chronic Chagas cardiomyopathy (CCC), no amastigotes nests are found in cardiac tissue, and it is now believed that sporadic recirculation of parasites may occur, perpetuating tissue damage and producing a sustained inflammatory response [9]. This hypothesis is based on detecting parasite-derived biomolecules (DNA, antigen) in cardiac tissue of the chronic phase [9].

Different models explain the persistence of the parasite: (1) immune alteration (by some immunosuppressive process) that can cause the parasite to replicate again, (2) latent forms of *T. cruzi* that evade the immune response with subsequent reactivation intermittently in replicative cycles, and (3) sporadic release of the parasite from distant organs (digestive tract), to recirculation and reinvasion of tissues [39].

#### 4.3.2. Parasite-Independent or Immune-Independent Myocardial Damage

During the acute stage, the release of parasites by rupture of infected cells stimulates a pro-inflammatory response with the activation of innate immunity mediated by *Natural Killer* lymphocytes, neutrophils, eosinophils, basophils, mast cells, and macrophages. These responses are maintained until the development of adaptive immunity (mediated by B, T CD4+, and CD8+ lymphocytes) responsible for eliminating the infected cells [9,54].

Non-specific antibodies are produced, and by the third week post-infection, specific antibodies are present in *T. cruzi* surface proteins, whose purpose is to destroy the parasite and control the infection. If the immune response produced is efficient, it will cause a reduction in the number of parasites. In most cases, *T. cruzi* manages to survive in the tissue and evades complement-mediated lysis and opsonization. To do so, it uses surface proteins that alter the binding between the initial molecules of the complement pathways and inhibit the formation of C3 convertase [40,55,56].

Paradoxically, the inflammatory response triggered for infection control will have caused progressive myocardial injury, characterized by vacuolization, myocytolysis, myofibrillar degeneration with subsequent fibrosis, and compensatory hypertrophy, key data for the chronic presentation of the disease [9,40,57].

In the indeterminate stage, the immune regulatory response associated with IL-10 and IL-17 predominates with high levels of Treg lymphocytes [9,9,58,59]. However, when Treg lymphocytes display deficient suppressor activity, the production of pro-inflammatory cytokines (TNF-α and IFN-γ) by Th1 cells is uncontrolled, creating an immune imbalance [2,49,60]. This inflammatory medium favors the appearance of microvascular phenomena that contribute to cardiac damage. Finally, all these mechanisms together perpetuate cardiac damage, progressing to the chronic symptomatic phase (clinical phase) [32].

During CCC, prolonged inflammation causes a delayed hypersensitivity response, characterized by inflammatory infiltrates composed mainly of T cells, macrophages, and destruction of myocardial fibers, associated with consequent myocellular hypertrophy and reparative fibrosis [49]. The latter causes a disorganization of the extracellular matrix, which, together with the hypertrophy and subsequent dilation of the heart, leads to cardiac remodeling and creates an environment conducive to the formation of arrhythmogenic foci, as well as alterations in contraction, with the consequent impairment of cardiac function [61,62].

Two mechanisms of autoimmunity in CD have been proposed: (1) the so-called *bystander activation* of T lymphocytes and (2) molecular mimicry [63,64]. The first is characterized by the release of self-antigens (against actin, myosin, muscarinic acetylcholine receptor, and cardiac muscle tubulin) and perpetuation of inflammatory phenomena secondary to exposure to intracellular proteins (parasite-dependent damage) [55,63]. Molecular mimicry, considered the most important mechanism of autoimmunity in CCC, refers to the similarity between parasitic and self-peptide sequences that provoke a cross-reaction and consequent production of autoreactive antibodies (autoimmune aggression of the muscle fiber) [64,65].

Although some studies correlate high titers of autoantibodies with the severity of CCC, other studies claim that such association does not exist. Thus, the relevance of these autoantibodies in cellular damage during CD is still controversial [66]. So far, only a small part of the actual role of autoimmunity in the clinical development of the disease is known.

#### 4.3.3. Microvascular Disorders

Microvascular disorders play an important role in the development of myocardial ischemia and myocardial fibrosis [14,55]. Endothelial dysfunction in patients with CD is associated with increased platelet activation, microthrombi, alterations in vasomotor control, and others, which predispose to thrombus formation in different vessels (cardiac, pulmonary, or cerebral) [14,55,67], as well as segmental alterations of myocardial wall mobility [68].

Nuclear medicine studies have found microvascular changes in the setting of chronic myocardial inflammation. The progression of left ventricular dysfunction is related to the extent and severity of myocardial perfusion abnormalities [68,69,70].

Clinical and experimental studies in murine models have demonstrated that individuals with CCC have abnormalities in microvascular perfusion regulation, microthrombi, and endothelial dysfunction [67,69,71]. The combination of these alterations could cause atypical chest pain; nevertheless, patients with confirmed CD may have angiographically normal coronary arteries, which provides an initial misdiagnosis [69,72,73,74].

At the molecular level, thromboxane A2 and endothelin-1 (ETE-1) have been related as pro-inflammatory agents, causing platelet aggregation and vascular spasms. On the one hand, thromboxane A2, produced by both the host and the parasite (90% attributed to the latter), facilitates platelet adhesion to the endothelium and cell proliferation and migration [75,76]. On the other hand, ETE-1, synthesized by infected endothelial cells and cardiomyocytes, causes vasospasm and contributes to endothelial dysfunction. Therefore, one might think that blocking the ETE-1 receptor would reduce the risk of these thrombotic events. However, in experimental studies in murine models, treatment with this ETE-1 receptor antagonist induced an increase in parasitemia (at the circulatory and myocardial levels) [77].

#### 4.3.4. Neurogenic Disorders

It has been suggested that nerve damage is the joint result of direct damage by the parasite, inflammation, and anti-neuronal immune reactions [78,79]. In patients with CD, there is a significant loss of neuronal cells of the autonomic nervous system, with the destruction of intracardiac autonomic neurons, mainly parasympathetic postganglionic neurons [80,81].

Inflammation compromises intercellular stimulus conduction by causing a decrease in intercellular communicating junctions and deposition of fibroblasts that prolong the action potential [40,62]. This electrical uncoupling results in slow stimulus conduction and unidirectional block, which, together with fibrotic areas, generate a reentrant circuit to form ventricular arrhythmias [55].

On the other hand, it is known that both parasympathetic and sympathetic vagal branches of the autonomic nervous system are also capable of modulating the immune response, mainly by shifting the predominance of a Th1 response toward a Th2 lymphocyte response [55,82]. It is believed that neurotransmitters released by these nerve endings can bind to specific surface receptors on immunocompetent cells that initiate regulatory mechanisms [55].

## 5. Clinical Presentation

The natural history of CD is diverse and multifactorial, as previously described, with a clinical spectrum and prognosis that depend on several factors (parasitic lineage, the patient’s immunological response, parasite and host genetics, and co-infections, among others) [16,80]. Two phases characterize it: acute and chronic, both with possible cardiovascular involvement but with a greater predisposition in the chronic stage (Figure 5) [81].

### 5.1. Acute Stage

On average, this stage lasts two months, with more severe effects in the first or second decade of life [85,86]. Clinical manifestations occur from day 8 to 10 after primary infection; 90 to 95% of those infected will be asymptomatic [85]. The main symptom is fever, accompanied by systemic symptoms such as general malaise, asthenia, adynamia, anorexia, headache, and arthralgias, among others [38,85].

On some occasions, it is possible to observe lesions related to the inoculation site, such as the Romaña sign (unilateral palpebral edema) or the chagoma (indurated lesion, elevated secondary to the bite, moderately painful) [1,39].

Cardiovascular manifestations are rare at this stage (~1%), but the most described are the presence of heart failure due to myocarditis, pericarditis, and conduction system abnormalities with a mortality of less than 5% [7,18,38,85,87]. The main causes of death are meningoencephalitis and myocarditis [52]. Sinus tachycardia, ST-T segment changes, PR or QT interval prolongation, QRS complex with decreased voltage, premature ventricular contractions, right bundle branch block (RBBB), or first-degree atrioventricular (AV) block are the most frequently described electrocardiographic alterations [38,85,86].

Laboratory findings are not specific (leukopenia, lymphocytosis, neutropenia, eosinophilia), and serological tests are generally negative (first weeks). Xenodiagnostic tests can observe circulating parasites; other detection methods are indirect immunofluorescence techniques or polymerase chain reactions (PCR) [86]. In the acute phase, it is rare to find alterations in imaging studies. Chest X-ray may reveal varying degrees of global cardiomegaly, or a transthoracic echocardiogram (TTE) may reveal pericardial effusion, tricuspid, or mitral insufficiency, concentric left ventricle (LV) hypertrophy. However, these alterations are found in less than 5% of cases [6,85].

Clinical manifestations resolve mostly spontaneously (90%) within two months, even without medical treatment [88].

### 5.2. Chronic Stage

In the chronic stage, parasitemia levels are undetectable. Approximately 30% of patients have cardiomyopathy with a high risk of progression to heart failure (HF) or fatal ventricular arrhythmias [14]. This stage is subdivided into two categories: chronic asymptomatic (also called indeterminate stage, subclinical, preclinical, latent, cardiac potential, or laboratory stage [89,90]) and chronic symptomatic or clinical (Figure 3).

#### 5.2.1. Chronic Asymptomatic Phase

The absence of visceral lesions characterizes this stage [85]. The following criteria are used to integrate the diagnosis: positive test for *T. cruzi* confirmed by serological or parasitological tests, normal electrocardiogram (ECG), no radiological alterations (chest, esophagus, or colon), and absence of clinical signs or symptoms of the disease [91].

Identifying this stage is challenging and often underdiagnosed due to the absence of symptoms. However, endocardial biopsies have been performed in animals and humans, demonstrating histopathological changes in inflammation, fibrosis, and myocardial degeneration compatible with myocarditis [14,92,93].

Regardless of being in an asymptomatic phase, it should be remembered that patients may serve as potentially infectious carriers and may eventually spread the disease by vertical transmission, blood transfusion, or organ donation [2,14]. Many patients remain in this stage for the rest of their lives (70%), and only 30% develop the symptomatic form, in which the most frequent presentation is dilated cardiomyopathy, with a progression rate of 1.85% to 7% annually [14,91,94]. On the other hand, cardiac fibrosis and inflammation are hallmarks of chronic Chagas cardiomyopathy (CCC) that can lead to sudden death [48,95].

#### 5.2.2. Chronic Symptomatic Stage

In general, this stage appears 10 to 30 years after infection [37,96,97]. The chronic symptomatic stage can be subdivided into three main forms: cardiac (20–30%), digestive (10–20%), and cardiodigestive (5–10%) [12,18,98].

The factors that determine disease progression or the onset of target organ damage are not known. Such disease progression has been associated with different variables such as genetic predisposition, geographical area, type of infection (vertical, oral, among others), immunosuppression status, concomitant chronic diseases, or some triggers such as age, male sex, alcoholism, the persistence of high parasitemia, the severity of the disease in the acute phase, among others [2,14,90].

On the other hand, the characteristic conditions of a chronic CD of the digestive system include colonic and esophageal disease, with alterations at the level of motility, secretory, or absorptive functions, both with intramural neuronal damage (Auerbach’s plexus) [12,85]. The esophageal disease can range from an asymptomatic motility disorder with mild achalasia to severe megaesophagus with dysphagia, odynophagia, chest pain, weight loss, and coughing, among others [2]. The colonic disease may progress to megacolon with chronic constipation and even to the manifestation of fecaloma, volvulus, or intestinal ischemia [12].

##### Chagas Cardiomyopathy (CC)

This is the most frequent type of manifestation (clinical phase); it can be divided into (1) conduction system abnormalities, (2) heart failure, and (3) thromboembolism (systemic or pulmonary) [85].

Heart failure secondary to CC can be classified according to the Latin American Guidelines for Diagnosis and Treatment of Chagas disease [99] in four stages: A, B (B1, B2), C, and D (Table 1), in which myocardial damage will be progressive and dilated cardiomyopathy will be the later and more severe manifestation of the disease [91].

## 6. Diagnosis

Diagnosis of CD is based on the detection of the parasite in the acute phase (high levels of parasitemia); there are three methods of parasite detection: (1) direct methods (confirmation of the presence of *T. cruzi*), (2) indirect methods, and (3) molecular tests [12,38]. In the chronic phase, there are serological methods, at least two of which must be used to confirm the diagnosis. More than 30 assays are available [12]**,** but the most commonly used are enzyme-linked immunosorbent assay, indirect immunofluorescence, and indirect hemagglutination [38,100].

Recently, simpler diagnostic methods are currently being researched to be used in the field, or in rural areas, without the need for sophisticated or high-tech equipment. The goal is to implement simple, low-cost, and affordable technologies with acceptable and validated performance, which complement the results of the central laboratory. This type of device is commonly named point–of–care (POC) [101]. A POC employing a nucleic acid amplification procedure has been developed, which shows a high sensitivity for *T. cruzi* detection; this is a potential diagnostic tool in humans for rural and low-income places, but the further investigation should be performed [102].

On the other hand, a molecular POC (RPA–LF, recombinase polymerase amplification-lateral flow) was developed to identify dogs infected with *T. cruzi*. This strategy can be well implemented in resource-limited areas, based on the detection of the canine reservoir (sentinel animals and reservoir host) [103]. Nowadays, the search for new elements to develop the diagnostic test with more sensitivity or specificity such as different chimeric antigens or biosensors, are the key to this purpose [104,105].

Likewise, other researchers are interested in the development of laboratory techniques that are capable of identifying antibodies produced by *T. cruzi* infection regardless of their genetic lineage or DTU so that these are a universal test that can provide an accurate result, both in a region endemic and in countries outside the American continent [105,106,107,108].

### 6.1. Biomarkers

Different cardiorenal biomarkers have been evaluated in patients with CCC at different stages, looking for their relationship with mortality, complications, and disease severity. Some markers are galectin-3 (Gal-3), neutrophil gelatinase-associated lipocalin (NGAL), soluble ST2 protein (sST2), cystatin-C (Cys-c), an amino-terminal fragment of brain natriuretic peptide (NT-proBNP), and high-sensitivity troponin T (hs-cTnT). All biomarkers except NGAL and Gal-3 have been associated with cardiac transplantation, use of left ventricular assist devices, and death, and the combination of NT-proBNP and hs-cTnT or sST2 is associated with an increased risk of death in patients with CCC [109,110].

### 6.2. Electrocardiogram (ECG)

It is an easily accessible and useful tool for initial diagnosis, stratification, and follow-up [111]; the sum of electrocardiographic abnormalities has been associated with an increased risk of HF, stroke, and even death [112].

The main alterations that can be found are conduction system defects such as RBBB, left anterior fascicle block (LAFB), non-sustained ventricular tachycardia (VT), atrioventricular block of variable degree, sinus bradycardia, atrial fibrillation (AF), and alterations in the ST segment and T-wave [12,113].

The presence of ventricular arrhythmias has been correlated with the degree of ventricular dysfunction, but they can also occur in patients with relatively preserved ventricular function [85].VT is common among patients with CCC, but the mechanisms of its sustained and nonsustained form are still unknown. The inferolateral LV scar is the primary source of reentrant circuits in sustained VT [85,114].

No finding is specific, but the greater the number of alterations identified, the greater the severity of myocardial damage and the greater the mortality [14]. As an example, Braggion-Santos et al. (2022) are the first to correlate electrocardiographic abnormalities with scar mass and LV dysfunction by cardiac magnetic resonance imaging, showing that the presence of isolated RBBB has a benign pathway, whereas the presence of LAFB and RBBB represented a serious cardiac involvement [115].

Hence, a routine ECG should be performed despite a previously normal ECG, as it is recognized as a marker of cardiomyopathy progression [14,116,117]. There are different *predictors of mortality* and severity in the ECG, such as the duration of the QTcmax interval and QT interval dispersion, the presence of pathological Q waves, T wave axis deviation, and episodes of sustained VT [118,119]. Twenty-four-hour Holter monitoring is also required to look for complex ventricular arrhythmias, sinus node disease, or atrioventricular conduction [14].

### 6.3. Echocardiogram

An echocardiogram is the initial and follow-up imaging study in most patients with suspected cardiomyopathy [120]. In the acute phase of Chagas disease, the most frequent finding is pericardial effusion. Mortality in the acute phase is rare (about 1%) and is mainly related to myocarditis [85]. In the above cases, the echocardiographic findings may be motility alterations and LV thickening with significant systolic dysfunction [121].

Nascimento et al. (2013) reported that LV diastolic dysfunction was found in the chronic symptomatic stage of CD, increasing its prevalence as the disease progresses from the phase without systolic dysfunction to the chronic phase with heart failure. Tissue Doppler was the best tool to observe the deterioration of diastolic function, as it is an independent predictor of adverse clinical events [114].

In the early stages (B1 or B2), the echocardiogram may demonstrate alterations of LV segmental mobility ranging from hypokinesia to aneurysm formation [2,121,122]. The cardiac regions frequently affected are the basal segments of the inferior and inferolateral wall and the apex, alterations that cannot be attributed to obstructive coronary artery disease. Segmental LV wall mobility abnormalities identify patients at risk for ventricular arrhythmias and disease progression [121].

The study of segmental mobility alterations is subjective, so the evaluation of deformation by two–dimensional speckle tracking (2DST) can become a useful tool in the early stages by identifying alterations in areas without apparent involvement [5,123]. The use of 2DST is less useful in patients with obvious segmental mobility disorders (Figure 6), but it still serves as a tool to assess the heterogeneity of systolic contraction (mechanical dispersion); these alterations are associated with ventricular arrhythmias independent of LVEF [122,124].

LV apical aneurysms are a pathognomonic finding [98,122] of CCC that may be useful for differential diagnosis with dilated cardiomyopathy (Figure 7). Another frequent presentation site is the inferolateral wall of the LV, although they are not exclusive to these sites [14]. The presence of aneurysms frequently accompanies disease progression to LV systolic dysfunction, where three-dimensional echocardiography and the use of contrast agents offer advantages for their evaluation [122,124]. In advanced stages of the disease, these are characterized by generalized hypokinesia and LV systolic dysfunction, the latter being a predictor of mortality in CCC [124].

The onset of LV diastolic dysfunction can occur early in the disease, even in asymptomatic forms of the chronic stage, with a prevalence of up to 10% [114]. Diastolic and systolic dysfunction coexist in all patients in the advanced stages of the disease. Right ventricular (RV) systolic dysfunction is a marker of poor prognosis.

During dobutamine stress echocardiography, segmental mobility disturbances can be induced in patients with CD without obstructive coronary artery disease [122,124]. Alterations in the microvasculature have been demonstrated to cause perfusion defects in these patients and are believed to be the origin of myocardial damage. The alterations in perfusion occur early in the disease and precede the appearance of alterations in segmental mobility at rest [124].

### 6.4. Cardiac Magnetic Resonance (CMR) Imaging

CMR makes it possible to identify myocardial fibrosis using late gadolinium uptake as a contrast agent [125]. Up to 20% of asymptomatic chronic-stage patients without LV segmental motion abnormalities have signs of fibrosis due to CMR.

The extent of myocardial fibrosis correlates with the severity of LV systolic dysfunction and the occurrence of ventricular arrhythmias [124]. Late gadolinium enhancement (LGE) can be transmural (44%), intramyocardial (32%), subendocardial (11%), epicardial (11%), or subepicardial [70]. The most frequently affected areas are the inferolateral wall and the apex (Figure 8). Transmural enhancement of two or more segments is a strong predictor of ventricular arrhythmias independent of other factors such as LVEF, age, gender, and extent of fibrosis. The extent of myocardial fibrosis has the potential to become an indication for implantable cardioverter defibrillator (ICD) in patients with CCC [122].

CMR offers advantages over echocardiography in the morphological and functional evaluation of the LV and RV because it is a tool with high spatial resolution, and the determination of volumes (LVEF/RVEF) does not depend on geometric assumptions [126]; it is also the best noninvasive tool for the evaluation of myocardial fibrosis or necrosis [127] and offers timely detection of RV systolic dysfunction in those patients in whom systolic dysfunction is not LV-dependent or when CC is not defined by clinical criteria [128]. Up to 3% of patients have LV apical aneurysms not detected by echocardiography [71,124]. However, the sometimes limited access to CMR preserves echocardiography as the initial and follow-up study of choice [124].

### 6.5. Nuclear Cardiology

Nuclear cardiology perfusion studies can reveal reversible or non-reversible defects that simulate infarcts in regions of myocardial fibrosis [129]. The study of cardiac sympathetic denervation by nuclear cardiology has promising uses in the early identification of CD and risk stratification of ventricular arrhythmias, but the lack of standardized protocols limits its use [72,124,130].

Cardiac tomography may be an option in patients with contraindications (allergies to gadolinium contrast medium or ferromagnetic medical implants not compatible with the machine in use [131]) or unavailability of CMR [122,124] to characterize LVEF, coronary anatomy, myocardial perfusion defects, and ventricular aneurysms (Figure 9).

## 7. Treatment

### 7.1. Medical Treatment

Effective first-line pharmacological treatment since the 1970s has included benznidazole (BZN) and nifurtimox (NF), each administered for a 60-day course of monotherapy [2,132,133].

BZN is considered the drug of choice and the best tolerated due to its high level of safety [7,12,134] and parasite reduction, but it does not reduce cardiac deterioration in the chronic phase [135]. There are studies where NF is better tolerated than BZN. In animal models, it has been shown that the susceptibility to these drugs is influenced by the type of *T. cruzi* strain present [136,137,138,139].

The main side effects of BZN are thrombocytopenic purpura, leukopenia, peripheral neuropathy, dermatitis, anorexia, weight loss, vomiting, or nausea; therefore, a complete blood count 21 days after starting treatment is recommended [14,18]. The main adverse effects of NF include systemic symptoms such as weight loss, anorexia, irritability, drowsiness, diarrhea, and vomiting; in general, all these effects are reversible, and only <1% are severe [133,140,141].

It is recognized that the highest efficacy rate of both drugs is during the acute and chronic asymptomatic phases, with a cure rate of 60% to 100% [12]. During the chronic phase, neither drug is effective, and they are contraindicated during pregnancy and renal or hepatic insufficiency [12].

In order to prove the efficacy of BZN in patients with CCC, the Benznidazole Evaluation for Interrupting Trypanosomiasis (BENEFIT) study was developed, in which 2854 patients from various countries in Central and South America recruited who received BZN or placebo for 80 days and were followed up for a mean of 5.4 years. In patients who were treated with BZN, the presence of parasites in serum was significantly reduced, but cardiac deterioration was not prevented through the five years of clinical follow-up [135]

Continuing with the clinical trials, in order to determine whether posazonazole (POS) alone or in combination with BZN was more effective than BZN monotherapy, as is usually used, 120 patients from Latin America and Spain were recruited. Trypnostatic activity of POS was demonstrated during treatment but did not demonstrate superior efficacy alone or in combination with BZN over BZN monotherapy [142].

Nowadays, there are several medications that are being tested under the drug repositioning strategy in order to avoid or reduce the clinical complications of CD. For example, fexinidazole, a medication used to treat sleeping disorders, has demonstrated effectiveness against the *T. cruzi* parasite in pre-clinical studies [143]. On the other hand, triazole derivatives, including ketoconazole, itraconazole, posaconazole, voriconazole, and ravuconazole –known asergosterol synthesis inhibitors– have been used for antifungal therapy and have shown to be suppressive but not curative against *T. cruzi* in humans and experimental animals [144]. Moreover, allopurinol, a medication typically used for treating gout, is an analog of hypoxanthine that prevents the synthesis of purines. Allopurinol and BZN administered together in mice during the acute phase were more effective than either treatment administered alone, according to a recent study that demonstrated a synergistic interaction of the drugs in vitro. Human clinical trials have been controversial. When used during the chronic phase of Chagas disease, allopurinol successfully delayed or even stopped the development of ECG abnormalities. However, it was also discovered that in patients with acute and chronic forms of Chagas disease, the medication was ineffective at inducing a parasitological cure [145,146,147]. Otherwise, there are alternative compounds, such as the oxaborole-containing substance AN4169, a nitroheterocyclic compound such as BZN and NF, that exhibited an effect against *T. cruzi* in vitro [148].

Other treatment approaches include the administration of immunomodulators with BZN or NF or even other antiparasitic drugs, as well as the introduction of alternative therapies with plant extracts. Several research has examined the effectiveness of plant extracts and their chemical derivatives against *T. cruzi*. In Colombia, some plants with possible antiparasitic effects have been tested against this parasite in vitro [149]. Numerous plant studies and their antiparasitic effects against *T. cruzi,* both in vitro and in vivo, have been conducted in Brazil and Argentina [150,151]. In Mexico, the seed extracts of tropical fruits such as papaya (*Carica papaya*) and avocado (*Persea americana*) have been examined. At a dosage of 250–500 μg/mL avocado extract demonstrated modest anti-epimastigote efficacy in vitro, while papaya fatty acids decreased the levels of both parasite stages (trypomastigote and amastigote) in mice [152,153].

The specific treatment of heart disease is focused on preventing sudden death, and among its main causes are fatal ventricular arrhythmias such as VT that degenerate into ventricular fibrillation [12]; so the use of ICD, antiarrhythmic drugs, or both are necessary [38]. In some cases, ablation of the reentrant circuit causing the tachycardia may be helpful; however, each case must be treated on a case-by-case basis [154].

In a double-blind, randomized study PARADIGM-HF trial, 8442 patients with heart failure and reduced ejection fraction were recruited, and the use of the angiotensin receptor–neprilysin inhibitor LCZ696, which consists of the neprilysin inhibitor sacubitril (AHU377) and the ARB valsartan, was compared with the use of enalapril, which had previously been shown to improve the survival rate of patients. According to this PARDIGM-HF trial, sacubitril/valsartan is effective in reducing deaths and hospitalizations for heart failure [155]. Its use in heart failure due to chagasic cardiomyopathy has not been extensively studied; there are reports showing a reduction in cardiovascular deaths and hospitalizations for heart failure, although with little statistical value [156,157]. Recently, in a prospective case series, the use of sacubitril/valsartan was analyzed after six months of use in Chagasic cardiomyopathy, showing improvement in symptoms according to the NYHA functional class, although without echocardiographic improvement. More studies are needed to identify its possible benefits in this disease [158].

### 7.2. Surgical Treatment

#### 7.2.1. Heart Transplant

A heart transplant is an alternative for patients with chronic HF refractory to medical treatment secondary to CC [20]. Survival per year has been estimated at 71% and at ten years at 46% post-transplant. However, there is a risk of disease reactivation secondary to post-transplant immunosuppression, although the evidence indicates that this could be treated with antimicrobial therapy as described above [159,160].

Acute cellular rejection appears to be the predominant complication in these patients [161]. There is a 70% rate of parasitic recurrence within the first year after heart transplantation, with a 10% mortality rate in chagasic recipients [162]. Some studies show that a reduced immunosuppressive regimen does not increase allograft rejection in patients and is associated with lower rates of CD reactivation; studies have been conducted where azathioprine/cyclosporine combination causes less reactivation compared to mycophenolate [160]. Currently, infection and rejection are the leading causes of death among CD heart transplant recipients, occurring in 21% and 10–14% of patients, respectively [161].

#### 7.2.2. Implantable Cardioverter Defibrillator (ICD)

Fibrosis, inflammation, and dysautonomia are involved in the genesis of ventricular arrhythmias in CCC [122], and the ICD is a prevention option for sudden death. There is no clear recommendation for early implantation in the current guidelines [163], and the treatment of ventricular arrhythmias is essentially empirical, based on recommendations extrapolated for heart diseases of other etiologies [154]. ICD placement should be indicated only when cost-effective and restricted to patients who can benefit from treatment, although the high-cost limits routine use [154].

## 8. Prevention

So far, there are only two drugs for the specific treatment of CD, but in the last decades, work has been performed on different types of immunological technologies for the creation of vaccines for both the prevention and treatment of the disease [164,165,166,167,168,169,170,171,172,173,174,175,176,177,178,179,180,181,182,183,184,185,186,187,188,189,190,191].

The development of an effective human vaccine against CD has faced many difficulties, as well as a slow process, due to the continuing controversy over its probable autoimmune etiology and the disinterest of the industry in supporting the development of this vaccine [162].

Immunological protection against *T. cruzi* has been studied since the second decade of the last century using animal models in which immunogens from killed parasites, attenuated parasites, cell fractions, purified proteins, and nucleic acid vaccines have been tested [162]. One of the most recent proposals for the development of a vaccine is the bioinformatics approach, which employs a wide range of computational techniques, including sequence and structural alignment, database design and data mining, macromolecular geometry, phylogenetic tree construction, prediction of protein structure and function, gene finding, and expression data clustering with the goal of functional analysis and data mining of data sets leading to biologically interpretable results and insights; recently this tool has been applied to the field of vaccinology against *T. cruzi* with promising results [192].

In both the acute and chronic phases, promising results have been obtained, such as reducing parasitemia and tissue parasite load, reducing inflammation, preventing electrocardiographic alterations and progression of the disease, and avoiding damage to cardiac tissue [164].

## 9. Conclusions

CCC is complex despite being a disease known for more than a century, and the World Health Organization currently classifies it as a Neglected Tropical Disease. The epidemiology of CD has changed due to migratory movements turning it from being typically endemic in Latin America into a global disease; therefore, more non-endemic countries are facing new waves of hospitalizations due to this challenging disease being ignored by physicians in their differential diagnoses.

The presence of non-ischemic cardiomyopathy in young adults with electrocardiographic abnormalities such as RBBB or AFB, atrioventricular block, sinus bradycardia, and premature ventricular contractions will make it necessary to rule out CD with complementary studies such as TTE, Holter monitoring, and serological tests. The presence of apical aneurysms, dilated cardiomyopathy, mural thrombi, and ventricular dysfunction are signs that should alert the clinician to suspect possible CCC. Given the current scenario, a multidisciplinary, comprehensive, and timely approach will lead to a positive outcome and a better quality of life for these patients.

CD is a disease with long-term clinical repercussions and a high cost of care that are exacerbated by the social and economic conditions of the inhabitants of endemic countries, where, despite the efforts of public health policies, its impact persists to the present day. Therefore, research on community prevention strategies (pharmacological or non-pharmacological) has become increasingly important in recent years.

## Figures and Tables

**Figure 1 jcm-11-07262-f001:**
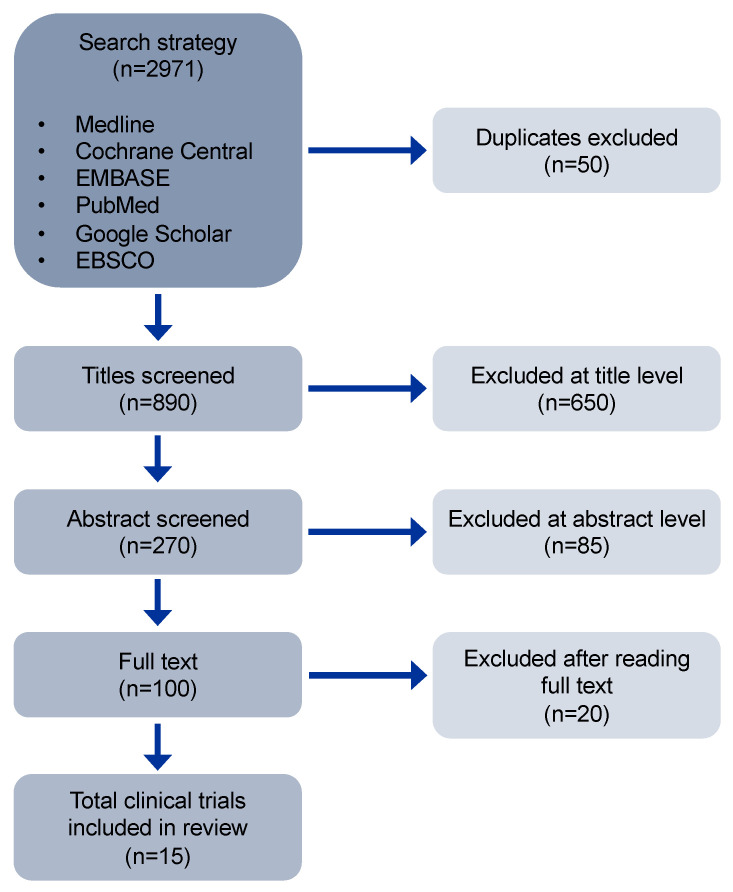
Flow diagram of selection process of eligible studies.

**Figure 2 jcm-11-07262-f002:**
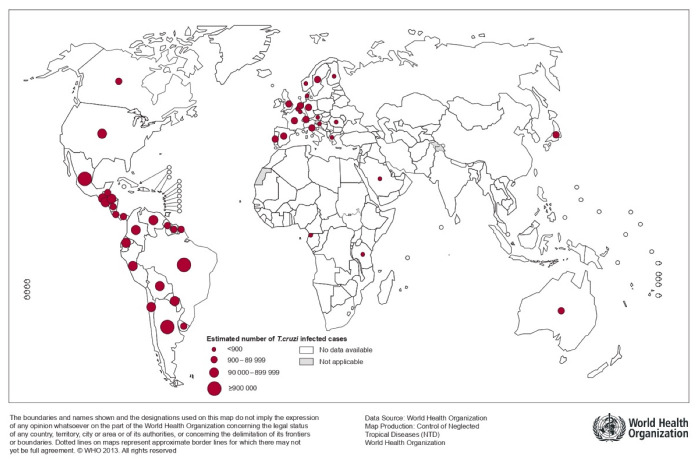
Global distribution of Chagas disease, based on official estimates, 2018 [11].

**Figure 3 jcm-11-07262-f003:**
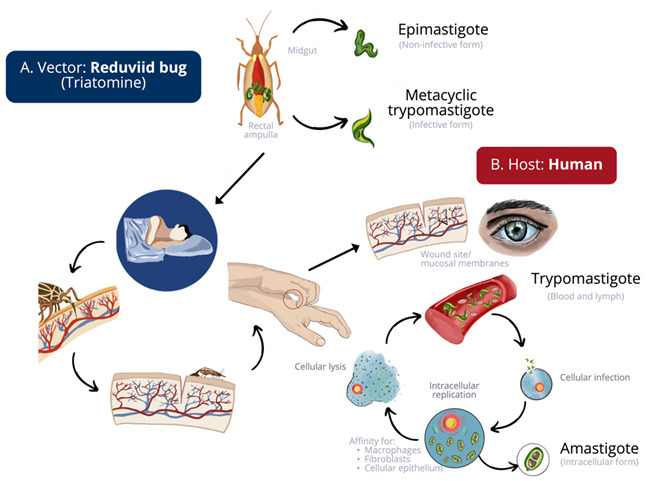
**Life cycle of *Trypanosoma cruzi.*** (**A**) Triatomines present two forms of the parasite: epimastigotes and metacyclic trypomastigotes. Epimastigotes will initiate their transformation to metacyclic trypomastigotes in the digestive tract of the triatomine (hindgut) after 8 to 10 days post-feeding. Trypomastigotes will gather in the rectal ampulla; during or shortly after blood ingestion, the kissing bug will defecate on the skin or near the mucous membranes (conjunctivae) of the vertebrate. (**B**) Entry of metacyclic trypomastigotes into the bloodstream, with subsequent infection of surrounding cells. Within the cells, they will transform into intracellular amastigotes, multiplying by binary fission. The amastigotes are able to lyse the cell and infect new ones or undergo differentiation into blood trypomastigotes and cause cell membrane rupture to disperse to other tissues. This cycle can be perpetuated, and the parasites can reinfect different host cells, with a predominance of muscle (myocardium) and neuronal cells [2,8,12,29,30,37,38,39,40].

**Figure 4 jcm-11-07262-f004:**
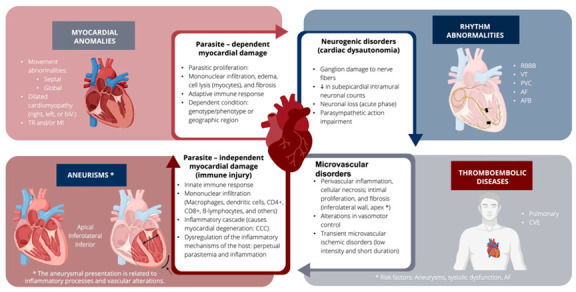
Correlation of pathophysiological mechanisms with the clinical manifestations of CD. RBBB: right bundle branch block, VT: ventricular tachycardia, AF: atrial fibrillation, PVC: premature ventricular contractions, AFB: anterior fascicular block, CVE: cerebrovascular event, TR: tricuspid regurgitation, MI: mitral insufficiency, RV: right ventricle, biV: biventricular.

**Figure 5 jcm-11-07262-f005:**
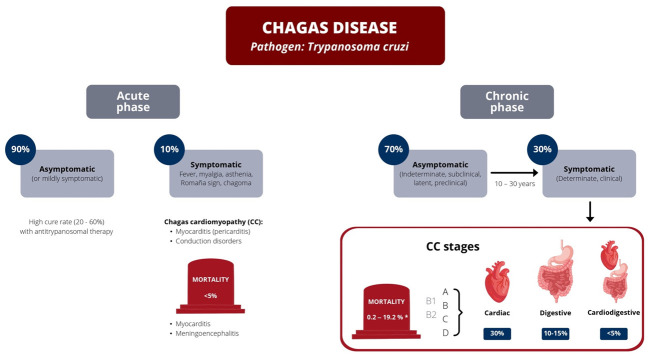
Clinical presentation and classification of heart failure secondary to *T. cruzi* infection are divided into four stages. * Annual mortality: a review of observational studies on predictors of mortality in chronic Chagas Disease [83]. Stages A, B (B1, B2), C, D are described in Table 1, according to the Latin American Guidelines for Diagnosis and Treatment of Chagas disease [82].

**Figure 6 jcm-11-07262-f006:**
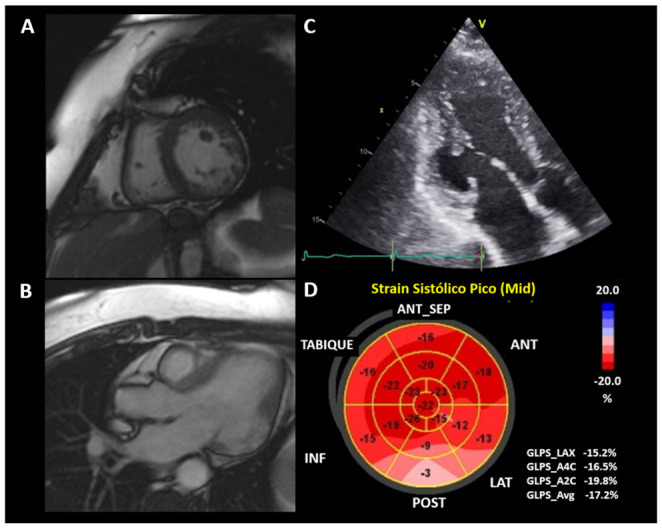
67-year-old woman was admitted to the emergency department for stable VT. (**A**,**B**) cardiac magnetic resonance shows LV lateral wall aneurysm in basal and middle thirds with late enhancement. (**C**,**D**) TTE(delate:-)–2DST shows the lateral aneurysm and the LV deformation. CD was confirmed by anti-*Trypanosoma cruzi* antibodies. Source: images taken at the *Instituto Nacional de Cardiología, Ignacio Chávez* (INCICh).

**Figure 7 jcm-11-07262-f007:**
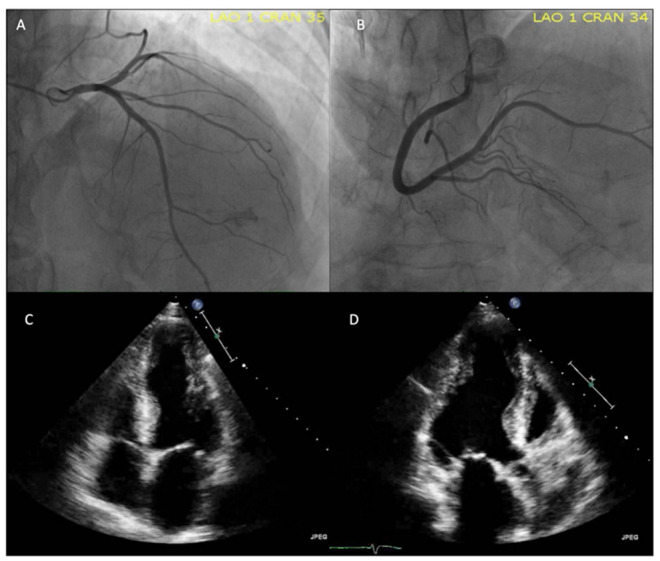
Coronary angiography and TTE study in a 74-year-old man with recurrent syncope and VT events who was diagnosed with CD due to the detection of anti-*Trypanosoma cruzi* antibodies and implanted with an automatic defibrillator. (**A**,**B**) Coronary angiography without evidence of lesions. (**C**,**D**) TTE with an apical aneurysm and another one in the lateral region of the LV in basal segments. Source: images taken at the *Instituto Nacional de Cardiología, Ignacio Chávez* (INCICh).

**Figure 8 jcm-11-07262-f008:**
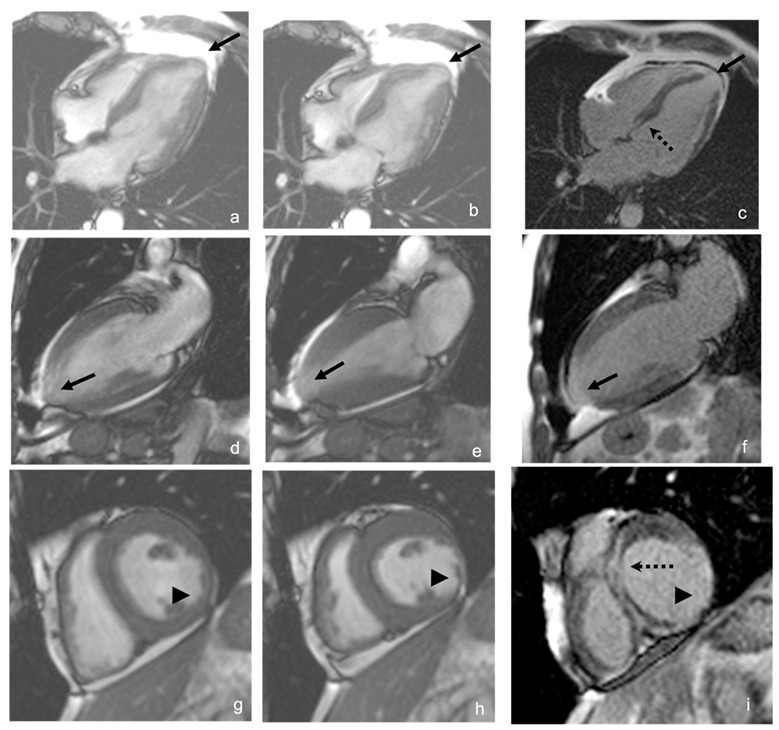
Cardiac magnetic resonance. Images in four chambers (upper row), two chambers (middle row), and short axis middle third (lower row). In the first column (**a**,**d**,**g**), images in diastole, the middle column (**b**,**e**,**h**) in systole, and the right column is inversion recovery sequence (**c**,**f**,**i**). Apical aneurysm (arrow), inferolateral thinning, and akinesia (arrowhead), with late transmural enhancement at this level, as well as a mid-wall septal late enhancement (dashed arrow), are shown. Source: images taken at the *Instituto Nacional de Cardiología, Ignacio Chávez* (INCICh).

**Figure 9 jcm-11-07262-f009:**
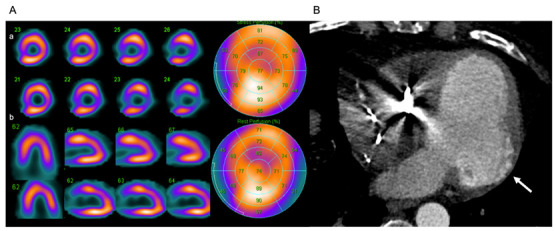
Cardiac tomography in a 66-year-old man diagnosed with CD by detection of anti-*Trypanosoma cruzi* antibodies. (**A**) Perfusion study by nuclear cardiology (using ^99m^ Technetium-methoxyisobutylisonitrile- ^99m^ Tc-MIBI-) shows non-transmural infarction of the anterolateral and inferolateral wall in its basal thirds without residual ischemia: (**a**) upper line stress phase and (**b**) lower line resting phase. (**B**) Coronary tomography angiography study without evidence of coronary lesions, with 0 Agatston units and basal inferolateral LV aneurysm (arrow). Artifice by implantable automatic defibrillator cable in the right ventricle is observed. Source: images taken at the *Instituto Nacional de Cardiología, Ignacio Chávez* (INCICh).

**Table 1 jcm-11-07262-t001:** Stages of development of heart failure secondary to *T. cruzi* [2,14,18,84].

A	B		C	D
Asymptomatic form (indeterminate): no data of cardiac or structural damage (normal ECG and chest X-ray).	Asymptomatic patients with structural or functional cardiomyopathy defined by electrocardiographic or echocardiographic findings		Symptomatic patients with data of previous or current HF. Significant LVEF impairment.	The presence of refractory symptoms secondary to HF, without response to treatment. NYHA (New York Heart Association) functional class IV. Specialized treatment
	B 1	B 2		
	Structural heart disease: mild changes by echocardiogram and ECG (arrhythmias or conduction disorders) Preservation of global ventricular function	Functional heart disease: decreased left ventricular ejection fraction (LVEF) Previously no signs or symptoms of heart failure		

## Data Availability

The data will be available from the corresponding author on reasonable request.

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
