# Peer review of "Chagas Heart Disease: Beyond a Single Complication, from Asymptomatic Disease to Heart Failure"

_jcm, 2022, doi:10.3390/jcm11247262_

Round 1

Reviewer 1 Report

The manuscript by Montalvo-Ocotoxtle and colleagues reviews the chagasic heart disease condition, focusing mainly on the pathogenic aspects and diagnostic features. The review is very well written, bibliographic information and data are well-curated and transcribed with logic and clarity in the final text.

The manuscript is clinical and biomedical importance, assembling relevant information to the field. However, I do have some minor comments and suggestions which I think should be addressed before accepting for publication. 

Specific/minor Comments

1) line 16: caused by the triatomine Trypanosoma cruzi (triatomine is the insect vector, replace the term by protozoan).

2)line 21 - clinical and diagnosis challenges are not restricted to non-endemic regions, they are also problematic in endemic areas. 

3)line 50: AT is used instead of CD, please uniformize the term for general use for better clarity. 

4)line 67: There are recent studies that acknowledge vectorial transmission within these USA regions. In my opinion, it is currently debatable to infer CC and CD incidence in the USA is solely associated with the Latin America migration process. The authors should add this information. Please check doi:10.4269/ajtmh.19-0733

5) line 73: In-built figure text (bottom left) is cropped. 

6) line 128 - fig2 legend - It is not clear which "time" authors mean by "they can lyse the cell at that time" - is it during amastiogte replication or during differentiation process? It should be better elaborated. 

7) line 175: the expression "by some toxic product released by the microorganism" is generic. Since the focus is on myocardial damage, authors should indicate some examples of molecules released by the parasite which are toxic for cardiac tissue. 

8) line 283 - figure 4 - why is the mortality (%) only indicated for acute phase and not for chronic phase? Also, it would also be graphic informative if author include in the figure the percenatge of infected individuals that will evolve to chronic phase. 

9) Line 414: author mention "mortality in the acute phase is rare (about 1%)" this is in disagreement with the percentage presented in figure 4 (about 5%). 

10)line 530: the authors describe cure rate of 60 - 100% of antiparasitic treatment, but it is not clear which drug they refer to (benzn. or nifurt.? or both?). 

Suggestions

1) As common in systematic reviews, authors should describe better their research methodlogy. The description is only available in the abstract, it is generic. An organigram detailing the total number of publication/data searched and which source (database) whoud be a valuable graphical and methodological addition to the manuscript. 

 2)Authors emphasize that socioeconomic impacts would also be approached, they mention the disease health and economic burden in developing countries. Nonetheless, when describing the CCC diagnostics (from simpler to more sophisticated techniques), as well as treatment and prevention, the description is merely technical. It would be important that authors address difficulties to implement such techniques in middle-and-low income regions, and the impact on underestimation of real cases, delayed  intervention, etc. 

3)Section "7" describes only perspectives of vaccines, using different approaches. The section is somehow disconnected with previous sections as authors never mention novel potential chemotherapeutic approaches (new drugs being developed - antiparasitic or pro-cardiac) or novel diagnostic methods. Authors could add more information on perspective of therapy, vector control and diagnosis to align with the detailed perspective text on immunology/vaccinology.      

Author Response

Dear reviewer:

We respectfully submit revision of our article, titled: Chagas Heart Disease: Beyond a single complication, from asymptomatic disease to Heart Failure, for your kind consideration. The article is submitted for MDPI Journal of Clinical Medicine.

We are thankful to you and another reviewer for providing the constructive feedback that has helped us to improve the presented contents. All points raised by the reviewers are addressed in this version, and we have provided point-by-point response to reviewer comments.

We look forward to hearing a positive response from you.

With warm regards

Gustavo Rojas–Velasco

Responsible of Cardiovascular Critical Care Unit,

Instituto Nacional de Cardiología Ignacio Chávez

Mexico City, Mexico

Reviewer 2 Report

Te review manuscript entitled "Chagas Heart Disease: Beyond a single complication, from asymptomatica disease to Heart Failure" is a well written and comprehensive manuscript, with few minor issues but that should be resolved:

> Some comments about the Benefit study and benznidazole and posaconazole should be added in treatment section

> Some comments about Chagas Disease subgroups analysis in sacubritil-valsartan trials should be added

Author Response

November 18th, 2022.

Dear reviewer:

We respectfully submit revision of our article, titled: Chagas Heart Disease: Beyond a single complication, from asymptomatic disease to Heart Failure, for your kind consideration. The article is submitted for MDPI Journal of Clinical Medicine.

We are thankful to you and another reviewer for providing the constructive feedback that has helped us to improve the presented contents. All points raised by the reviewers are addressed in this version, and we have provided point-by-point response to reviewer comments.

We look forward to hearing a positive response from you.

With warm regards

Gustavo Rojas–Velasco

Responsible of Cardiovascular Critical Care Unit,

Instituto Nacional de Cardiología Ignacio Chávez

Mexico City, Mexico

RESPONSE TO REVIEWERS

All changes were made using “Track Changes” function in Microsoft Word as Assistant Editor suggested. These modifications are in red color.
